# A Review on Electrochemical Microsensors for Ascorbic Acid Detection: Clinical, Pharmaceutical, and Food Safety Applications

**DOI:** 10.3390/mi14010041

**Published:** 2022-12-24

**Authors:** Totka Dodevska, Dobrin Hadzhiev, Ivan Shterev

**Affiliations:** Department of Organic Chemistry and Inorganic Chemistry, University of Food Technologies, 26 Maritza Blvd., 4002 Plovdiv, Bulgaria

**Keywords:** microelectrodes, sensors, electroanalysis, vitamin C

## Abstract

Nowadays, micro-sized sensors have become a hot topic in electroanalysis. Because of their excellent analytical features, microelectrodes are well-accepted tools for clinical, pharmaceutical, food safety, and environmental applications. In this brief review, we highlight the state-of-art electrochemical non-enzymatic microsensors for quantitative detection of ascorbic acid (also known as vitamin C). Ascorbic acid is a naturally occurring water-soluble organic compound with antioxidant properties and its quantitative determination in biological fluids, foods, cosmetics, etc., using electrochemical microsensors is of wide interest. Various electrochemical techniques have been applied to detect ascorbic acid with extremely high sensitivity, selectivity, reproducibility, and reliability, and apply to in vivo measurements. This review paper aims to give readers a clear view of advances in areas of electrode modification, successful strategies for signal amplification, and miniaturization techniques used in the electroanalytical devices for ascorbic acid. In conclusion, current challenges related to the microelectrodes design, and future perspectives are outlined.

## 1. Ascorbic Acid: Biological Role and Commercial Applications

L-Ascorbic acid (AA), also known as vitamin C, is an important target for quantitative sensing and monitoring in the biomedical field, food quality control, pharmaceutical, and cosmetic industries. AA plays an important physiological role in cellular redox metabolism by participating in free radical scavenging and immune system boosting. Antioxidant activity of AA helps to prevent certain diseases such as cancer, hypertension, neurodegenerative diseases, age-related muscular degeneration, cataracts, etc. [1,2]. AA is involved in the physiology of the nervous system, including the support and the structure of the neurons, and the processes of differentiation, maturation, and neuronal survival [3]. AA is a co-factor for at least 15 enzymes involved in the biosynthesis of collagen and L-carnitine, peptide hormone activation, tyrosine metabolism, synthesis of norepinephrine from dopamine, etc. [4].

As the human body is unable to synthesize AA endogenously, it is important to include adequate AA intake in the diet. AA is also used as a nutritional supplement during deficiency states (weakened body immunity, anemia, scurvy). Oral vitamin C produces tissue and plasma concentrations that the body tightly controls. Approximately 70–90% of vitamin C is absorbed at moderate intakes of 70–180 mg/day. High levels of AA are maintained in cells and tissues, and are highest in brain, pituitary gland, and adrenal glands. Relatively low levels of AA are found in extracellular fluids, such as plasma, and saliva. Side effects are not observed from normal AA intake, due to the fact that it is quickly excreted [5]. At doses above 1 g/day, absorption falls to less than 50% and absorbed, unmetabolized ascorbic acid is excreted in the urine. After single oral doses of vitamin C greater than 2 g daily, gastrointestinal distress and diarrhea are the most common side effects. However, in individuals with renal dysfunction, increased risk of kidney stones is observed as a serious side effect with vitamin C overdose. Here it should be noted that people most commonly use vitamin C for preventing and treating the common cold although the results are inconsistent and research is still ongoing in this field.

AA is widely used in dermatology and the cosmetic industry as a skin conditioning agent. Numerous clinical studies support the use of topically applied AA for photoprotection, anti-aging, and skin-lightening purposes [6,7,8,9]. The function of AA as an antioxidant and an enzyme cofactor is essential in maintaining skin health and preventing skin aging. Vitamin C is a cofactor of prolyl hydroxylase and lysyl hydroxylase, which are enzymes responsible for the hydroxylation of lysine and proline, a key step of collagen biosynthesis. In addition to stabilizing the collagen molecule by hydroxylation, vitamin C also stimulates collagen mRNA production by fibroblasts [10].

The topical AA treatment of the epidermal surface increases the synthesis of several specific lipids of the skin, enhancing the protective barrier function [11,12]. Vitamin C also provides protection against UV-induced photodamage and participates in the repair of oxidative DNA lesions in skin cells [13]. A topical vitamin C suppressed UVB-induced cell death, apoptosis, DNA damage, reactive oxygen species (ROS) production, and the inflammatory response by down regulating tumor necrosis factor-α (TNF-α) expression and release [14].

Vitamin C is a major component of commercially available cosmetics for the treatment of skin pigmentation disorders such as freckles, age spots, and melasma. Such dermatological problems can be caused by various factors such as hormonal imbalance during pregnancy or menopause, thyroid disease, side effects from certain medications, genetic predisposition, etc. Emerging evidence has indicated that Vitamin C has therapeutic effects on facial hyperpigmentation, as it reduces melanin synthesis. AA suppresses the catalytic activity of tyrosinase, the rate-limiting enzyme in melanin biosynthesis [15]. Although the transdermal absorption efficiency of AA is low and its anti-pigmentary and skin-protective mechanisms still need to be clarified, AA has been used widely as a skin-lightening, anti-aging, and anti-inflammatory agent in commercially available cosmetics designed to protect and rejuvenate photoaged skin.

The strong antioxidant activity of AA and its ability to protect oxidizable constituents, including phenolic and flavor compounds, is the main factor for AA to be frequently used as an additive in the food industry to prevent unwanted changes in color and flavor [16]. AA is used as color fixative, preservative, and acidity regulator in foods such as meat products, bakery products, canned fruit, canned meat, beer, jam, sweets, and fruit juices. Used as a preservative, AA can reduce the risk of mold and other microbial growth, thereby preventing food spoilage and browning.

As an electron donor, AA serves as one of most important low-molecular-weight antioxidants which contributes to the total antioxidant capacity—an important quality indicator of foods and drinks [17]. Since natural vitamin C is characterized by low thermal stability and a tendency to oxidize easily, each process using elevated temperature causes loss of this vitamin compared to fresh material. These losses can be from 20 to even 90% depending on the temperature level, the duration of the processing operation, and contact with oxygen [18]. Because vitamin C is easily destroyed by pressing and heat treatment, manufacturers of juices and fruit/vegetable purees use AA as an additive to either renew or improve the overall nutritional value of the product. Otherwise, the specific flavor of AA enhances the taste of some food products. Thus, fruit juices, jams, and candies often benefit from the tinge of acidity that provides the consumer with the diverse taste of fresh fruits.

Concerning the importance and wide application of AA, the reliable quantitative detection of AA became an important issue not only for the routine chemical analysis, but also for clinical biology, and the pharmaceutical and cosmetic industries. Monitoring AA content should be regarded as an essential and relevant task for evaluating the quality of food products, raw materials, and various pharmaceutical formulations, considering the nutritional value and therapeutic AA properties.

## 2. Electrochemical Microsensors for Ascorbic Acid Determination

The electroanalytical methods have demonstrated advantages over conventional methods such as titrimetric analysis, spectrophotometry, and liquid chromatography, demonstrating higher sensitivity, exceptional selectivity, remarkable repeatability and accuracy, low power requirements, operational simplicity, and cost effectiveness. More importantly, electrochemical sensor devices can be miniaturized, eventually enabling in vivo and in situ detection of the target analyte in the actual microenvironment [19]. Recent advances in microfabrication, surface modification, and signal processing have aided the design of highly sensitive and efficient microsensors.

It is well known that under micro (or nano) scale sizes electrodes show unique electroanalytical behavior compared with conventional electrodes. As a consequence of the reduced capacitive charging currents and increased mass transport rates, microelectrodes exhibit excellent signal-to-noise (S/N) characteristics, rapid response, and high-speed measurement. The high mass transfer rates allow electroanalytical measurements to be made at low substrate concentrations. Electrochemical microsensors enable samples to be analyzed in real time at the point of care or through implantation. Such an approach is vital for effective diagnoses, monitoring responses to treatment, and thus providing prognoses for chronic medical conditions [20]. In the last few years, research has convincingly shown that in vivo electrochemistry is one powerful strategy for probing brain chemistry. In vivo monitoring of the dynamics of the changes in neurochemicals becomes more and more important to unveil and understand brain activity and function [21].

Ascorbic acid is a ketolactone with two ionizable hydroxyl groups. pK_1_ is 4.2 and pK_2_ is 11.6, thus the ascorbate monoanion is the dominant form at physiological pH, while in acidic medium (pH < 4.2) AA remains in the protonated form. A generalized mechanism of the electrocatalytic oxidation of AA to dehydroascorbic acid, is shown in Figure 1. The oxidation of AA includes the transfer of two electrons and 2H^+^ ions, to produce dehydroascorbic acid, which was proven to be followed by an irreversible solvation reaction at pH lower than 4.0. This irreversible reaction yields an electroinactive product (2,3-diketogulonic acid) easily adsorbable on the electrode surface, which can result in electrode fouling [17].

Chronoamperometry and voltammetric techniques—cyclic voltammetry (CV), differential pulse voltammetry (DPV), and square wave voltammetry (SWV) (Figure 2) can be used as an efficient alternative, providing an affordable and accurate approach for fast quantitative determination of AA. Amperometry is based on the application of a constant potential to a working electrode, and the subsequent measurement of the current generated by the oxidation or reduction of an electroactive analyte. The resulting steady-state current is proportional to the bulk concentration of the analyte. Voltammetry is a potentiodynamic technique, based on measuring the current arising from oxidation or reduction reactions at the electrode surface when a controlled potential variation is imposed.

In the DPV technique, short pulses (10–100 ms) with limited amplitude (1–100 mV) are superimposed on a linear ramp. The current is immediately measured before the pulse application and at the end of the pulse; the difference between the currents is recorded and plotted versus the potential. The procedure effectively reduces the capacitive current, due to the direct current ramp. Accordingly, DPV shows a higher sensitivity and selectivity compared to CV due to the enhanced discrimination of Faradaic currents.

In SWV, a symmetrical square-wave pulse is superponated to a staircase wave. The duration of the pulse is equal to the length of the staircase, and the superponation is obtained in such a way that the forward pulse of the square wave coincides with the first half of that staircase. The first current is measured at the end of the forward square-wave pulse, and the second one is measured at the end of the return square-wave pulse; the signal is obtained as an intensity of the resulting differential current. The change in current between potential steps, is plotted versus the potential. SWV is a fast and powerful technique with extremely low detection limits comparable with those of chromatographic techniques.

The emergence of microsensor devices has provided new directions to various applications including disease diagnosis, pharmacology, and food safety. Depending on the application field (biomedical, pharmaceutical, or food safety), there are specific requirements to the sensor’s operational characteristics. In regard to the linear dynamic range, microsensors are required to detect the analyte in the actual nano-, micro-, or millimolar levels. Limits of detection and quantification also should be adequate to the lowest possible levels of the analyte in the particular sample. In terms of selectivity, the sample matrices (biofluids, tissues, foods, etc.) contain numerous different and specific molecules that have similar behavior to the target analyte and undergo electrochemical conversion. Therefore, the methodology of the selectivity experiments must adequately reflect the intended microsensor mechanism and application.

The next parts of the review present the most recent and innovative works published on microelectrodes and their application for AA detection in biological, pharmaceutical, and food samples, mostly focused on original studies reported from 2015 to date.

### 2.1. Clinical and Pharmaceutical Analysis

AA levels are closely correlated with physiological and pathological events in brain diseases, thus alterations to the AA concentration in the live brain might have diagnostic value for neurodegenerative diseases [23]. The cortex, hypothalamus, and cerebrospinal fluid (CSF) exhibit high concentrations of AA. At normal conditions in rat brain, AA is present in CSF at a concentration of about 200–500 μM, while blood levels are around 60 μM, highlighting an active role of AA uptake in the brain. According to the work of Zhang et al., the overall AA level was decreased below 200 μM in striatum and hypothalamus in the rat brain model of Alzheimer’s disease [23]. In humans, CSF ascorbate concentrations tend to be slightly lower at 160 μM (plasma values are 40–60 μM). Despite fluctuations of plasma ascorbate levels, the brain maintains high levels of AA, and CSF concentrations have been shown to remain relatively stable in healthy individuals. However, the mechanism still remains unclear, mainly due to the difficulty of accurately detecting AA levels in complicated live tissue. The high oxidation overpotential (around 300 mV at most types of electrodes), sluggish electron transfer kinetic value, the difficulties in obtaining good reproducibility of the electrode signal in direct electrochemical oxidation of AA, as well as the problems of electrode bio-fouling make selective electrochemical measurements of AA in vivo a longstanding challenge. There are a number of interferences coexisting in cerebral systems; several electroactive species (metal ions, neurotransmitters, etc.) can generate interference responses over that of the target analyte, leading to difficulty for accurate in vivo sensing in live brains [24].

The most investigated selectivity problem is the interference between AA and dopamine (DA)—one of the most important catecholamine neurotransmitters in the mammalian central nervous system. The abnormal levels of DA are associated with neurological disorders such as schizophrenia, Parkinson’s, and Alzheimer’s diseases. Carbon-based materials are typically selected as sensing electrodes when AA or neurotransmitters are desired to be measured. However, the values of the oxidation potentials of AA and DA are very similar at bare electrodes. To overcome this problem, various strategies (chemical functionalization, nanomaterial doping, surface treatments, conducting polymer coatings, and molecularly imprinted polymers) have been developed and adopted in order to separate the oxidation potentials of DA and AA. Common surface modification procedures include the application of specifically selected materials that inherently separate the oxidation potentials of DA and AA. Modified microelectrodes based on quantum dots, nano-sized particles, and polymers, show remarkable selectivity and efficiency in AA detection. In this review, we tried to select the most representative articles to illustrate the main strategies used to develop AA-reliable electrochemical microsensors.

The variable surface morphology of carbon materials permits a variety of surface functionalities for the development of highly efficient and long-term stable electrochemical microsensors. Recently, carbon fiber (CF), carbon nanotube (CNT), and graphene-based microelectrodes have become research topics owing to their numerous advantages. As one of the most popular microelectrodes, the conventional carbon fiber (CF) microelectrode has been employed for in vivo quantitative determination of AA because of its high conductivity, excellent biocompatibility, and ease to be miniaturized. According to the studies of Cao et al. [24], AA oxidation at a CF microelectrode was observed with a high overpotential, indicating a slow electron transfer process (CV data registered in neutral medium). This result may be explained by the inactivated surface of the CF microelectrode resulted from the electrode fouling by the oxidation product of AA. The same research group developed a new electrochemical approach for detection of cerebral AA in rat brain microdialysates based on a novel electrochemically pretreated nitrogen-doped carbon nanotube fiber (e-NCNF) microelectrode. The presented e-NCNF retained excellent electrochemical catalytic properties for AA oxidation at a low potential (0.0 V vs. Ag/AgCl), exhibiting remarkable selectivity, high sensitivity and good electrode-to-electrode reproducibility. The e-NCNF electrode exhibited good electrocatalytic activity; it had a linear current response in the concentration range from 50 to 1 mM, sensitivity of 1.1 mA mM^−1^ cm^−2^, a detection limit of 20 µM, and good selectivity. Due to avoiding the manual modification procedures, such NCNF essentially provided a facile way for microelectrode preparation and greatly improved the reproducibility of electrode fabrication. The deviation of anodic current intensity of six electrodes prepared with same method did not exceed 4% (RSD, *n* = 6). This strategy essentially simplifies the modification process and thus minimizes the person-to-person and electrode-to-electrode deviation. Based on the remarkable analytical performance, the e-NCNF with good biocompatibility was successfully applied to determine the AA level as 134 ± 7 mM in rat brain microdialysates. Generally, the commented work offers a new electroanalytical platform for fabricating engineered microelectrodes applicable for in vivo measurements.

Carbon nanotubes (CNTs) have been of great interest in electrochemical sensor design due to their remarkable mechanical and electrical properties: large specific surface area, fast charge transfer, excellent compatibility with other materials, and strong synergistic effect. Electrocatalytic activity (capability to increase the electron-transfer kinetics and therefore to reduce the overpotential of redox reaction) towards a number of electrochemical processes of analytical significance, as well as the unique stability are key properties of CNT-based electrodes that are widely used in electroanalysis. CNTs are classified primarily according to their shape and size as single-walled carbon nanotubes (SWCNTs) or multiwalled carbon nanotubes (MWCNTs). CNTs efficiently facilitate the electrochemical oxidation of AA, establishing novel minimized and easily prepared platforms for sensitive and selective AA detection [25].

The use of vertically aligned carbon nanotube-sheathed carbon fibers (VACNT-CFs) as pristine microelectrodes for real-time, in vivo monitoring of ascorbate was reported by Xiang et al. [26]. Authors have observed a significant acceleration in ascorbate oxidation on the electrodes electrochemically pretreated in NaOH solution, suggesting that the electron transfer of ascorbate may be facilitated at the oxidized and opened tips of CNTs. The VACNT-CFs were formed via pyrolysis of iron phthalocyanine on the CF support, followed by electrochemical pre-treatment in 1.0 M NaOH solution (Figure 3). Such an approach successfully avoided the manual electrode modification procedures and thus minimized deviations intrinsically associated with conventional CF electrode fabrication, which often involves electrode surface modification with randomly distributed CNTs or other pretreatments. Using the as-synthesized VACNT-CFs as microelectrodes without any post-synthesis functionalization, the authors have been developing a new strategy for in vivo monitoring of ascorbate with high selectivity and reproducibility. In this study, for in vivo electrochemical measurements, a tissue-implantable micro-sized Ag/AgCl electrode was used as a reference electrode. The reference electrode was prepared by first polarizing Ag wire (1 mm diameter) at +0.6 V in 0.1 M hydrochloride acid for ca. 30 min to produce an Ag/AgCl wire and then inserting the as-prepared Ag/AgCl wire into a pulled glass capillary, in which aCSF was sucked from the fine end of the capillary (o.d. ca. 30 μm) and used as the inner solution.

Ferreira et al. demonstrated that nanocomposite sensors consisting of carbon fiber microelectrodes (CFMEs) modified with Nafion^®^ and CNTs, display electrocatalytic properties towards the oxidation of ascorbate [27] and suitable recording properties in vivo (selectivity, sensitivity, and minimal biofouling). Authors have used SWCNTs/Nafion^®^ modified CFME to estimate in vivo basal ascorbate levels in the sub-regions of the rat hippocampus. Furthermore, ceramic-based Pt microelectrode arrays coated with glutamate oxidase were used to selectively monitor potassium-evoked glutamate release and reuptake in the rat hippocampus with high spatial and temporal resolution. Finally, the two microbiosensors were assembled in an array (Figure 4) and used for simultaneous real-time measurement of ascorbate and glutamate in vivo to assess the dynamic interplay between these two neurochemicals.

In order to increase the specific surface area and conductivity of the CFs, CFMEs are modified using various carbon materials, metal/metal oxide nanoparticles, metal organic frameworks, etc. Methods such as electrodeposition, chemical vapor deposition, electroless plating, self-assembly, and drop casting were used successfully to enhance the electrode surface area, number of the active sites, and to regulate the functional groups. Several high-quality review articles cover the progress in methodologies and approaches for effective modification of CFMEs [28,29,30,31].

To increase both the material conductivity and specific surface area, Dong et al. have applied a simple, rapid, and highly purified radio frequency (RF) magnetron sputtering-assisted hydrothermal method to prepare a carbon fiber/ZnO coaxial nanocable microelectrode (CF/ZnO CN-ME) to detect AA [32]. Although the authors did not present results from in vivo studies, the study provides a portable and green route for the utilization of nanomaterials in clinical diagnosis.

The adsorption of macromolecules, in particular proteins, on the surface of microelectrodes during in vivo electrochemical measuring inevitably leads to a decrease in the sensitivity. In 2016, Zhang’s group developed the protein pre-treated strategy to modify the electrode surfaces to stop protein adsorption. They have reported that the pre-treatment of carbon fiber microelectrodes with bovine serum albumin (BSA) offers a simple strategy to minimize the interference from the alternation of proteins in the brain [21]. A high reliability was achieved using this method for the electrochemical in vivo measurements of dopamine and ascorbate. The authors concluded that: (1) the adsorption of BSA onto CFMEs by immersing the electrodes in an artificial cerebral spinal fluid (aCSF) containing BSA substantially validates a new and effective pre-calibration method for in vivo measurements by adding BSA into aCSF, and (2) the use of BSA-treated CFMEs as a probe for in vivo measurements virtually eliminates the variation of current responses caused by the alteration of proteins during electrode implantation and following brain activities.

Recently, the effectiveness of the protein pre-treated strategy for ascorbate monitoring was validated using a gelatin-MWCNTs modified carbon fiber microelectrode [33]. The gelatin-MWCNTs/CFME after the BSA pre-treatment exhibited excellent biocompatibility, stable sensitivity, and selectivity for amperometric detection of ascorbate in aCSF as well as in diluted blood samples.

Ratiometric electrochemical sensors (RECSs) have attracted extensive attention, since these devices are able to overcome the system errors of the conventional electrochemical sensors derived from the alteration of the environment and operating personnel [34,35]. The RECS possesses dual electrochemical signals, and the quantitative detection of the target is based on the ratio of these two signals. The ratiometric signal readout mode provides a built-in correction factor to eliminate the contribution from non-specific interferences [34]. With this strategy, the RECSs were endowed with enhanced accuracy, reproducibility, and reliability, which is a highly desirable method applicable for in vivo repetitive analysis. To date, the research on RECSs for AA determination was still in its infancy, and only few reports have been published.

Zhang et al. in 2017 presented a novel aligned carbon nanotube fiber for the ratiometric detection of AA levels in live rat brains with Alzheimer’s disease [23]. Engineering tunable defects and oxygen-containing species in CNTs, the research group developed a ratiometric electrochemical microsensor that provides a fast, selective, and accurate method for detecting AA levels in live brains. The oxygen levels were controlled using two strategies—oxygen plasma treatment (o-CNF) and electrochemical polarization (e-CNF). The results clearly demonstrated that the presence of oxygen-containing groups remarkably increased the electron transfer kinetics for AA oxidation. On the DPV, a new oxidation peak at −270 mV (vs. Ag/AgCl) was obtained for the e-CNF microelectrode, whereas no peak was observed for o-CNF. The new peak is related to the redox process of the quinone groups produced after electrochemical treatment; it was completely separate from the peak of AA oxidation (−50 mV) and its density remained unchanged as the AA concentration increased. Therefore, it can effectively serve as an internal reference to provide a built-in correction that avoids the environmental effects of the complex real system. The ratiometric peak current density (*J_p_*/*J_p_*^0^) was linear to the AA concentration in the range of 50 µM to 1 mM. The developed e-CNF-based microsensor demonstrated a low limit of detection (10 µM) and high sensitivity (0.85 mA M^−1^ cm^−2^). The low potential for AA oxidation at the pretreated e-CNF microelectrode is critical for determining AA levels with high selectivity in cerebral systems. No obvious responses (<2.0 %) were observed upon the addition of neurotransmitters and other potential interfering compounds. Data for electroactive neurotransmitters including dopamine (DA), 5-hydroxytryptamine (5-HT), uric acid (UA), and 3,4-dihydroxyphenylacetic acid (DOPAC), show that their peak potentials were more positive and well separated from that of AA. The e-CNF-based ratiometric microsensor was used successfully to detect AA in vivo in a living normal rat brain and in a rat brain model of Alzheimer’s disease (Figure 5). The e-CNF microelectrodes were implanted in different regions in the rat brain according to standard stereotaxic procedures. Another 2 mm plastic cannula was located at 5 mm far from the working electrode, in which reference and counter electrodes were introduced. High electrode-to-electrode reproducibility and good anti-biofouling ability make the e-CNF microelectrode a promising candidate for in vivo monitoring of AA levels.

Cheng et al. reported a ratiometric electrochemical microsensor for effective and reliable detection of AA in living brains [36]. The sensor was constructed by immobilizing preassembled thionine/Ketjen black (KB) nanocomposites (with typical size around 50 nm) onto carbon fiber microelectrodes. As an electrode substrate the authors chose carbon black (Ketjen black, KB), a promising carbon nanomaterial that possesses good conductivity, high surface to-area ratio, and low cost. The KB efficiently facilitated AA oxidation at a relatively negative potential (−0.1 V) without particular physical or chemical pretreatment, forming the basis of selective measurement of AA. With a well-defined and reversible pair of redox waves at −0.21 V, thionine (Th, assembled onto KB via hydrophobic and π−π stacking interactions) acted as an internal reference. The in vitro experiments demonstrated that the sensor exhibited extremely high reproducibility and stability toward selective measurement of AA. Here it should be noted that in vivo performance of the developed ratiometric electrochemical sensor has some peculiarities—the recorded redox waves of thionine were broadened compared with CVs of in vitro experiments. The observed feature might be ascribed to severe electrode fouling as well as enhanced resistance in cerebral systems. The designed sensor was successfully applied in vivo to effectively, selectively, and reliably monitor the dynamic change of cerebral AA associated with pathological processes in living rats’ brains. The authors have provided data on consecutive CVs in vivo recorded at a low scan rate every 5 min at a thionine/KB-modified CFME implanted in the cortex of a rat brain. The CVs remained unchanged with a stable I_AA_/I_Th_ ratio during a long-run monitoring, suggesting the good stability and reproducibility of the ratiometric electrochemical sensor for endogenous AA measurements.

For the first time, Jiang et al. in 2020 have reported a facile ratiometric electrochemical sensor for in vivo/online repetitive measurements of cerebral AA in brain microdialysate [37]. Methylene blue (MB) was electrostatically adsorbed onto the graphene oxide (GO) surface as a built-in effective internal reference to achieve ratiometric detection of AA. The authors have demonstrated that the electroreduction of GO facilitated the AA oxidation; the electron-transfer kinetics of AA oxidation were markedly enhanced at the ERGO-MB electrode. After the electroreduction of GO, the overpotential for AA oxidation (DPV mode) was reduced by 278 mV at the ERGO-MB electrode and the peak anodic current was also increased by 311%. A well-defined oxidation wave at a formal potential of −428 mV (MB) remained nearly the same with the occurrence of AA oxidation (−100 mV), indicating that the assembled MB did not take part in the ERGO-facilitated oxidation of AA (Figure 6). The in vitro experiments demonstrated that the as-designed RECS exhibited high reproducibility, good selectivity, and high sensitivity. The ratio between the two oxidation peak currents (I_AA_/I_MB_) revealed an excellent dynamic range for AA concentration at 0.5–1000 μM. The detection limit was as low as 10 nM, which is superior to most of the previously reported AA sensors. Such a dynamic range well covered the physiological AA concentration of cerebral microdialysate (2–12 μM). The developed RECS was successfully applied in the electrochemical detection−microdialysis system to in vivo/online repetitive monitoring of the dynamic change of cerebral AA effectively and reliably in the progress of the global cerebral ischemia/reperfusion events.

Dong et al. developed a novel electrochemical sensor by molecular design for the simultaneous detection and quantification of HClO and AA in body fluids [20]. A latent MBS electrochemical molecular probe was designed and used as a HClO-specific recognition molecule. Anthraquinone (AQ) was used as an internal reference compound. The CFME/ERGO−CNT/AQ+MBS microelectrode exhibited strong electrocatalytic activity for AA oxidation. The electrochemical platform was used to monitor AA and HClO in human body fluids (saliva, urine, serum).

Thin-film technologies enable the use of a metal (non-inks) surface as classical solid electrodes, but with a low-cost and high fabrication resolution. Deposited metallic thin films exhibit superior mechanical properties that would be further enhanced by forming a nanocomposite in the thin film layer. Taking advantage of their inherent properties such as extremely high sensitivity, low reagent consumption, being reusable, as well as having non-tedious pre-treatment procedures, thin-film microelectrodes are a useful tool for enhancing the electroanalytical parameters in multiple applications. A gold interdigitated microelectrode modified by graphene oxide decorated with gold nanoparticles (AuNPs-GO/Au-IDA) has been developed for simultaneous determination of AA and uric acid in human urine samples [38].

Transition metal hexacyanoferrates (MeHCF) have received considerable attention in material science and the electrochemical sensing field due to their unique characteristics–easy and cheap synthesis, good stability, ability to mediate/catalyze electrochemical reactions, and fast kinetics [39,40]. MeHCF deposits as electrode modifiers have been extensively investigated to exploit their electrocatalytic properties towards various organic and inorganic species. The usefulness of ruthenium oxide hexacyanoferrate RuOHCF-modified CFNT carbon fiber disc (CFD) microelectrodes (r = 14.5 μm) for in vivo monitoring of ascorbate inside neuroblastoma cells was demonstrated by Paixão et al. [41]. Ogorevc and coworkers have reported the development of a microsensor prepared by modifying the surface of a substrate carbon fiber microelectrode with an electrochemically deposited nickel oxide and ruthenium hexacyanoferrate (NiO–RuHCF) sensing layer, using a two-step galvanostatic/potentiodynamic protocol, with further application of a cellulose acetate membrane (CAM) protective coating [42]. The proposed microsensor was found to be suitable for measuring AA under strong acidic conditions (pH 2.0), applying hydrodynamic amperometry at a constant potential of 0.3 V (vs. Ag/AgCl, sat. KCl). The calibration plot obtained in the extremely complex matrix of real undiluted gastric juice was linear from 10 to 520 μM. The authors concluded that the presented microsensor has a promising potential for direct real-time amperometric sensing of AA at physiological levels in real gastric juice environments, exhibiting prolonged stability.

Nowadays, nanostructured metal oxides are considered to be one of the most fascinating functional materials owing to their outstanding catalytic/electrocatalytic/optical properties. In electrochemical sensor design, a variety of metal oxide nanostructures (nanorods, nanowires, nanorings, etc.) have been extensively utilized as efficient electrode substrates for developing new-generation nanodevices with high performance. Ruthenium dioxide (RuO_2_) is a promising candidate as an electrode sensing material in electrochemical devices due to its high electric conductivity, extremely low resistivity, excellent chemical stability, and electrocatalytic property. The electrocatalytic activity of RuO_2_-nanowires grown on electrospun TiO_2_ nanofibers for the oxidation of AA was investigated by Kim et al. [43]. Electrocatalysts are based on TiO_2_-nanofiber skeletons, of which surfaces are decorated with RuO_2_ nanowires of enlarged surface area. Thus, the use of TiO_2_-nanofiber backbones can greatly increase the surface-to-volume ratio of the RuO_2_ material while preventing the self-agglomeration of RuO_2_ nanowires. All the electrochemical characteristics verified high sensitivity, low detection limit (<1.8 μM), good reproducibility, and reasonable stability, suggesting the feasible use of the developed electrode material for AA sensing in real biological samples. The applications of RuO_2_ nanowires and TiO_2_ nanofibers as an efficient sensing platform were confirmed for the quantification of AA in commercial vitamin C tablets, vitamin beverage samples, and for the real-time direct analysis of AA generated from living rat liver tissue in vitro without additional electrode modification or sample separation steps.

The fine control of nanoporous gold (NPG) characteristics allow the wide-range tuning of its electrical and electrocatalytical properties [44]. The sensitivity to AA of conventional Au microelectrodes was increased about one thousand-fold upon modification with NPG film, according to Kumar et al. [19]. A selective and robust NPG-modified gold microelectrode-based electroanalytical platform was designed for the detection of AA in acidic extracts (pH 1.0) of *Aspergillus fumigatus* fungus and *Arabidopsis thaliana* leaves. The obtained results were in good agreement with those found by spectrophotometric measurements.

An acupuncture needle (ANE) is used in Eastern medicine for medical therapies. Recently, there is strong evidence that micro-needle electrodes based on stainless steel ANEs have promising applications in constructing electrochemical sensing platforms. Jia et al. in 2019 developed a non-enzymatic AA microsensor on the basis of ANEs functionalized with a Ni_6_MnO_8_ nanoflake layer [45]. Authors have employed a simple hydrothermal reaction–calcination process for the growth of Ni_6_MnO_8_ nanoflakes on an ANE coated with a carbon film. Ni_6_MnO_8_ shows a large specific surface area (Figure 7) and exclusive electronic characteristics. The modified electrode possesses high ability and feasibility for AA determination. Under the optimized conditions, an extremely high sensitivity of 3106 μA mM^−1^ cm^−2^, a broad linear range between 1.0 μM and 2.0 mM, and LOD = 0.1 μM for AA were defined. Furthermore, to demonstrate the feasibility of the prepared sensor, the amounts of AA in vitamin C tablets were tested. The recoveries determined for AA ranged between 96.5 and 99.3%, demonstrating its practicability and reliability.

Microneedle-based platforms could provide unprecedented real-time information about the patient’s status and enable early medical actions. In 2021, Cuartero et al. published a review highlighting the state of electrochemical microneedle sensors during the last decade [46]. This high-quality article deals with the important advancements in electrochemical microneedle sensor technology, among which are included new microneedle fabrication methods and various modification strategies, providing different architectures and allowing for the integration of electronics.

### 2.2. Non-Invasive Skin Analysis

Nowadays, non-invasive healthcare technologies have become increasingly important issues for researchers, clinicians, and patients. Wearable wireless sensor networks are being currently developed for real-time non-invasive monitoring of electrolytes and metabolites in sweat, tears, or saliva, providing an attractive alternative to the conventional medical care systems.

Non-invasive continuous AA monitoring methods are highly advantageous and desirable, as readings could be obtained without any patient discomfort. Conductive metal–organic frameworks (MOFs) are a class of porous materials consisting of organic nodes connected by organic linkers. These materials have many advantages, such as an excellent conductivity, simple synthesis protocol, adjustable structure, etc. MOFs provide unique well-defined porous structures, large pore volumes, many catalytically active sites, and high crystallinity, and so they are becoming increasingly important and attractive as electrocatalytic materials. Until recently, the integration of MOFs with flexible electronic devices for wearable sensing was a challenge.

Yang et al. in 2022 presented a wet-adhesive epidermal sweat sensor for metabolite detection by integrating an electrically conductive Ni-MOF (Ni_3_HHTP_2_) as the active layer with a flexible and breathable nanocellulose substrate (Figure 8) [47]. The developed robust sensor can conformably self-adhere to sweaty skin and exhibits high water-vapor permeability. A wireless epidermal nutrition tracking system for in situ monitoring of the dynamics changes in sweat vitamin C levels during daily activities C was demonstrated and the results were comparable to those of HPLC.

An enzyme-free, sensitive, portable system for AA detection was subsequently developed by integrating Ni_3_(HITP)_2_/SPCE as a sensing element, Bluetooth 5.0 hardware, a smartphone, and an application-specific integrated circuit (ASIC) to facilitate electrochemical data acquisition and analysis [48]. This portable sensor showed good electrocatalytic performance to accurately monitor the AA level in sweat as a part of personal health management. Commented results provide promising examples of MOF-based sensors for the non-invasive and continuous/periodic detection of nutritional balance.

A flexible sensor based on single-step modified Au microelectrodes for the electrochemical detection of AA in sweat was reported by Ibarlucea et al. [49]. The modification procedure consists of electrodeposition of biocompatible alginate membrane with entrapped commercially available CuO NPs. The electrode was fabricated using a thin polyimide support, and the soft nature of the membrane can withstand mechanical stress beyond requirements for skin monitoring.

Cristophe et al. fabricated (Pt–Pt–Ag/AgCl) and (Au–Pt–Ag/AgCl) electrochemical microcells (ElecCell) using silicon-based technologies derived from microelectronics in order to detect AA and uric acid [50]. The fabricated microdevices were tested for non-invasive skin analysis in order to determine the global antioxidant capacity detection feasibility. ElecCell microdevices (Au–Pt–Ag/AgCl) were applied without any pre-treatment directly on the forearm (Figure 9), using the “stratum corneum” natural hydrolipidic film to guarantee the electrical contact between the working (Au), counter (Pt), and reference (Ag/AgCl) microelectrodes.

Selection of materials and technique for microelectrodes development depends on the properties required for real applications. Various types of microelectrodes with different layouts, materials, and fabrication processes have been presented. At the end of this section we could summarize the strategies in fabrication of microelectrodes for AA detection as follows: (i) direct electrodeposition; (ii) modification; (iii) assembly techniques; (iv) screen-printing technique; (v) photolithography.

### 2.3. Food Analysis

To the best of our knowledge, there are a limited number of published articles dealing with the determination of AA in food samples by means of non-enzymatic microelectrode devices. A microelectrode made from a pyrolytic graphite sheet (PGS) was developed and tested for the quantitative detection of vitamin C in biological samples prepared from raw arugula (*Eruca sativa* L.) cultivated in soil with different types of “green” biowaste [51]. The electroanalytical determination was performed by square wave voltammetry, and the results for the obtained arugula samples planted on different substrates agree well with the classical determination methods at lower AA concentrations in the sample. However, at higher AA content in the sample, the results were slightly higher, compared to the standard method (redox titration using iodine). The authors report that the deviation is more pronounced at higher concentrations of AA.

A number of macrocyclic compounds (porphyrins, phthalocyanins) are suitable for attributing specific reactivity of sensing electrodes. Stefan-van Staden et al. have used different porphyrins for the design of carbon paste and diamond paste-based microelectrodes, which were employed for the determination of AA in pharmaceutical (Supradyn, Vitamin C tablets) and beverage (Prigat) samples using differential pulse voltammetry [52]. The researchers pointed out that the interference of dopamine in the response of most of the microelectrodes made their utilization for the assay of AA in biological samples not possible.

Thin films of polyaniline containing the dopant ions polyvinylsulfonate (PVS) and polystyrenesulfonate (PSS) have been prepared on 25 μm Pt disk microelectrodes and tested for direct AA determination in wine and orange juice-containing solutions [53]. A potential of 0.1 V (vs. SCE) was found to be most suited for the amperometric determination of AA in beverages diluted with a neutral pH buffer, minimizing the interference of oxidizable compounds such as polyphenols, glucose, catechin, and caffeic acid. However, the authors have reported that exposure to wine and orange juice led to a diminution of the sensor response, likely due to adsorption of unknown beverage components onto active polymer redox sites.

A platinum wire microelectrode was corroded using aqua regia and subsequently embedded with carboxyl functionalized MWCNTs to achieve more active sites, producing a so-called powder microelectrode (PME) [54]. A two-electrode electrochemical system was constructed to detect AA based on its direct oxidation on the electrode surface using the PME and a platinum plate electrode as a reference electrode. The proposed sensor displays a linear range of 5.0–950 μM (LOD = 4.89 × 10^−7^ M). Practical sample analysis (lemon, apple, vitamin C tablets) revealed the dual-electrode sensor system exhibited good accuracy, specificity, and reproducibility. A recovery rate of 94–107% with relative standard deviation (RSD) < 5 % has been reported.

In summary, the very limited number of microsensors successfully applied to quantify AA testifies that the food sample matrices still present significant challenges. Foods and beverages are complex, heterogeneous mixtures that naturally vary in constitution and pH value. Food samples containing a variety of low- and high-molecular weight compounds: proteins, lipids, sugars, polyphenols, pesticides, pigments, oxidizable acids (citric acid, caffeic acid, vanillic acid, 4-hydroxybenzoic acid, etc.), and bases (amines, etc.). Most of these substances are electroactive and capable of producing an interfering current signal. Some compounds cause electrode passivation (biofouling) forming an impermeable layer on the electrode surface, thus hindering the direct contact of the analyte with the electrode surface. In particular, at a suitable applied potential, phenolic compounds form a polymeric film on the electrode surface, thereby decreasing sensor signal.

## 3. Conclusion and Outlook Remarks

Recent advancements in the field of nanomaterials have greatly benefitted the electrochemical sensors’ features. Electrochemical techniques combined with micro- and nano-sized electrode materials have demonstrated rapid, low-cost, and reliable performances. Electrochemical microsensors allow clinical analysis with real-time monitoring capability, along with extremely high sensitivity, and selective detection of target biomolecules. In this regard, in vivo analysis based on electrochemical sensors has attracted outstanding interest in the interdisciplinary research fields spanning material science, nanotechnology, electrochemistry, and neuroscience.

This review article has highlighted the strategies that have been introduced to successfully improve the operational parameters of electrochemical microsensor devices as powerful tools for AA detection in clinical, biochemical, pharmaceutical, and food analysis. The results show that microsensors, based on various electrochemical techniques, detect AA with extremely high sensitivity and reliability, and were successfully applied to in vivo measurements. Although great achievements have been highlighted in applications of AA microsensors, there are still challenges to overcome:Microfabrication strategies have to address the problems related to the quantitative analysis of AA in real samples, emphasizing selectivity and reproducibility.Surface functionalization of microelectrodes to enhance analyte detection and reduce biofouling, thus increasing the operational stability.Engineering new electronic interfaces and analytical approaches that will improve the signal-to-noise ratio and increase the electrochemical signals generated by the target analyte.Using microelectrode arrays to build multianalyte sensing systems will be useful for diagnostic and therapeutic monitoring.Research aimed at implementing the modern advanced microsensor technologies in the food industry.

## Figures and Tables

**Figure 1 micromachines-14-00041-f001:**
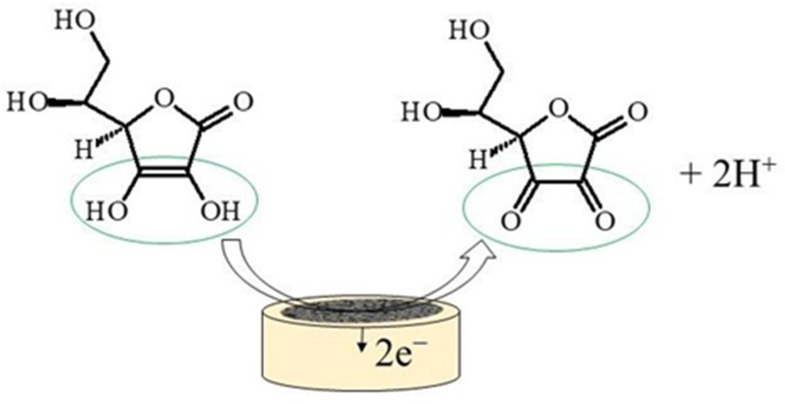
Electrocatalytic oxidation of ascorbic acid.

**Figure 2 micromachines-14-00041-f002:**
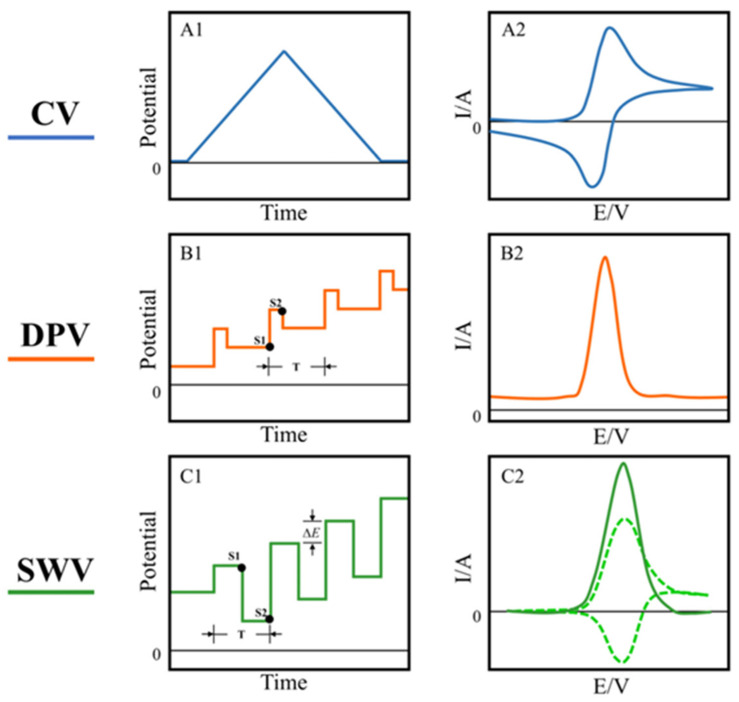
The potential (**A1**) and typical response current (**A2**) of CV; the potential waveform (**B1**) and voltammogram (**B2**) of DPV. In the potential waveform, T is the waveform period, and S1 and S2 are the two current sampling points; the typical potential waveform (**C1**) of SWV, ∆*E* is the potential increment, T is the potential period. The response current consists of forward (anodic current) and reverse (cathodic current) components (dashed line in (**C2**)), and their difference results in a net current (solid line in (**C2**)). Reproduced from Ref. [22]. Licensee 2022 MDPI.

**Figure 3 micromachines-14-00041-f003:**
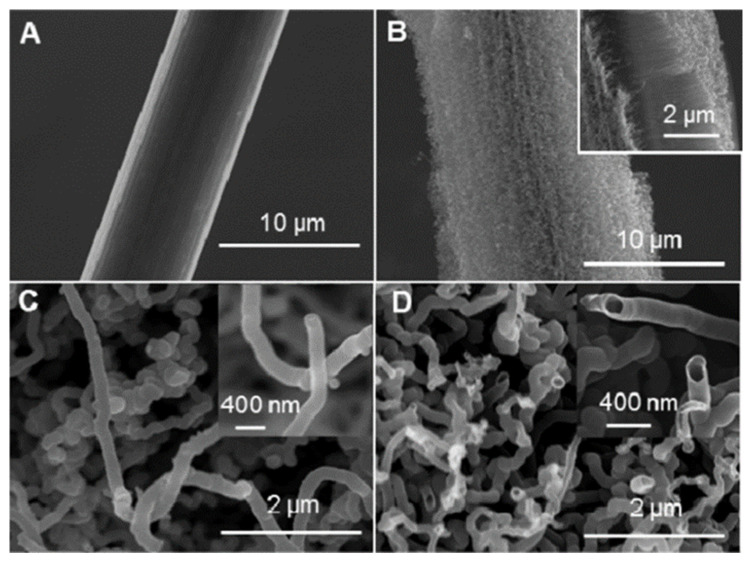
SEM images of (**A**) CF, (**B**) VACNT-CF, and the tips of VACNT-CF (**C**) before and (**D**) after electrochemical treatment in 1.0 M NaOH. Reprinted with permission from [26]. Copyright 2014 American Chemical Society.

**Figure 4 micromachines-14-00041-f004:**
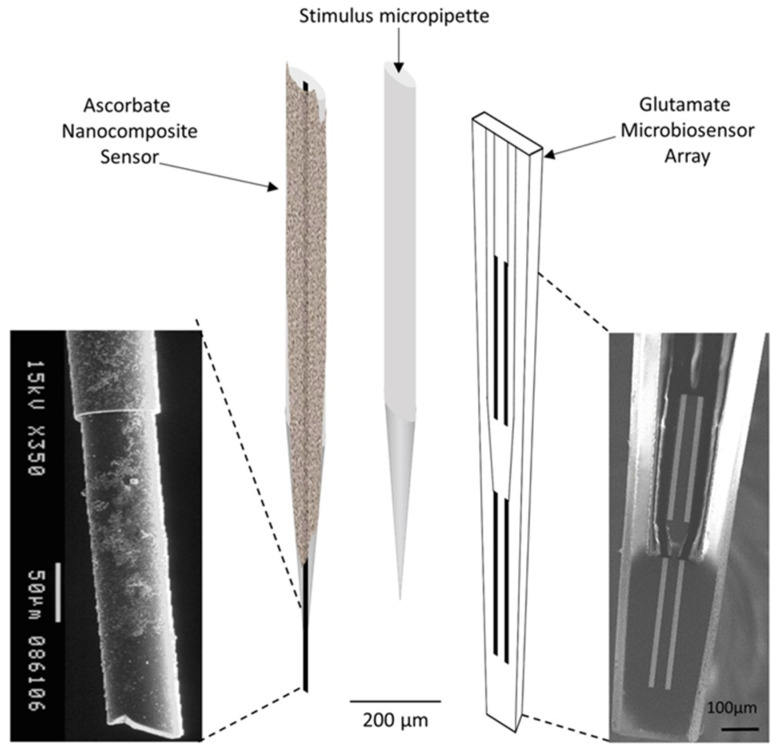
Schematic representation of the array composed by the ascorbate nanocomposite microsensor (**left**), the glutamate microbiosensor (**right**), and the micropipette (**center**) used for local application of solutions in the extracellular space of the rat hippocampus. Reproduced with permission from [27]. Copyright 2018 Elsevier.

**Figure 5 micromachines-14-00041-f005:**
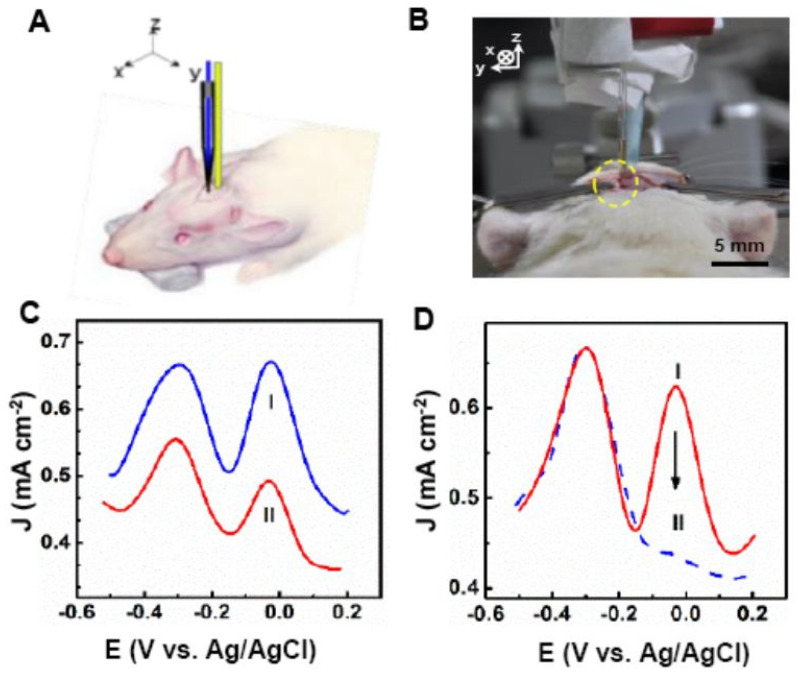
(**A**) Schematic illustration of the in vivo setup for determining AA in rat brain. (**B**) Optical images before and after the stereotaxic implant into the brain. (**C**) DPVs recorded at the e−CNF microelectrode in the striatum of a normal rat (I) and the rat brain model of AD (II). (**D**) DPV responses recorded at the e−CNF microelectrode in the striatum of the rat brain model of AD before (I) and after (II) injection of ascorbate oxidase. Reproduced with permission from [23]. Copyright 2017 American Chemical Society.

**Figure 6 micromachines-14-00041-f006:**
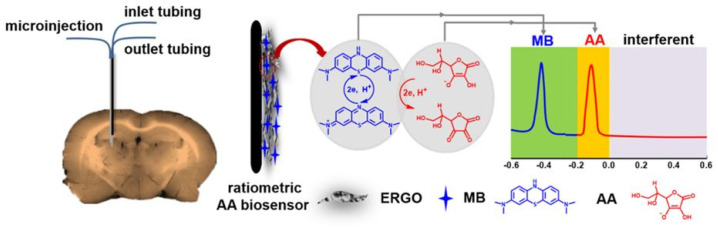
Schematic illustration of in vivo microinjection and the designed RECS for selective measurement of cerebral AA in brain microdialysate. Reproduced with permission from [37]. Copyright 2020 American Chemical Society.

**Figure 7 micromachines-14-00041-f007:**
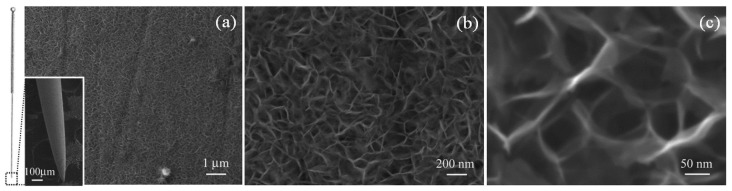
(**a**) Low-magnification SEM image of Ni_6_MnO_8_ nanoflakes; inset: SEM image of the modified acupuncture needle. (**b**,**c**) High-magnification SEM images of the Ni_6_MnO_8_ nanoflakes. Reproduced from Ref. [45] with permission from the Royal Society of Chemistry.

**Figure 8 micromachines-14-00041-f008:**
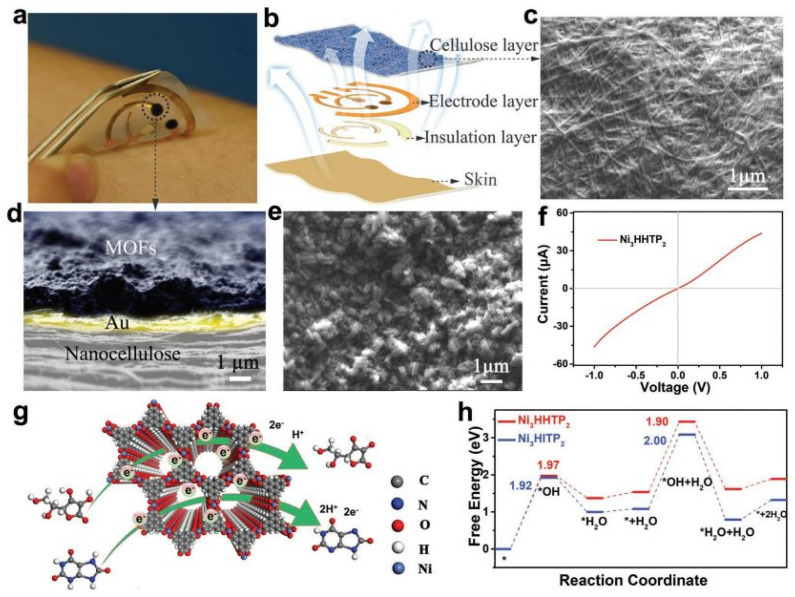
Design and structure of cMOF−based wearable sweat sensors. (**a**) Photograph of the wearable sweat sensor that self-adheres to the sweaty skin of humans. (**b**) Configuration of the layered film sensor, which comprises cMOF, electrode, cellulose, and insulation layers (Ecoflex was used as an insulating layer to encapsulate the interconnected area and avoid contact of the conductive interconnected path with the skin and perspiration). (**c**) SEM image of the prepared BNC membrane. (**d**,**e**) Sectional view and top view SEM images, respectively, of the Ni_3_HHTP_2_−based working electrode. (**f**) Current–voltage characteristic of Ni_3_HHTP_2_ measured using the two-contact probe method. (**g**) Schematic illustration of the oxidation mechanism of AA and UA catalyzed by Ni_3_HHTP_2_. (**h**) The free energy diagrams for AA electrocatalytic oxidation on Ni−based cMOFs using DFT calculations. Reproduced with permission from [47]. Copyright 2022 John Wiley and Sons.

**Figure 9 micromachines-14-00041-f009:**
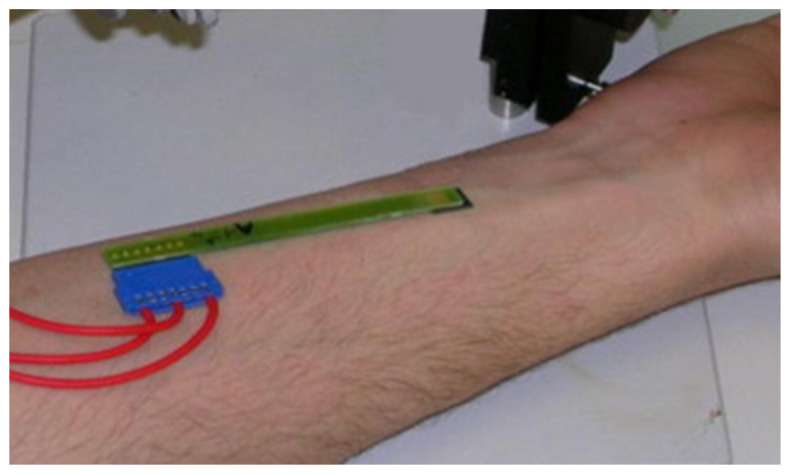
Use of the (Au–Pt–Ag/AgCl) electrochemical microcell for the skin analysis. Reproduced with permission from [50]. Copyright 2013 Elsevier.

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
