# Peer review of "A Review on Electrochemical Microsensors for Ascorbic Acid Detection: Clinical, Pharmaceutical, and Food Safety Applications"

_micromachines, 2022, doi:10.3390/mi14010041_

Round 1
Reviewer 1 Report
This review has finalized the development of electrochemical microsensors to detect AA. The biological role and commercial application of ascorbic acid were described and the electrochemical methods for its assessment have been reviewed from 2015 to date. Several main classes of AA sensing applications were analyzed. A systematic discussion of the selected results is presented from the point of reproducibility and stability toward AA measurement.
Some considerations and questions are listed below:
o Page 3, line 120: concerning electrochemical oxidation of AA, add literature data on ascorbic acid oxidation at different pH and possible products since the coexistence of species produces from AA electrochemical oxidation influences its selective detection.
o Page 3, line 135: explain in detail the basic principles of DPV and SWV methods.
o Page 4 line 149: АА levels are closely correlated with physiological and pathological events in brain… What are the physiological АА concentrations?
o Page 7 lines 240 and 242: add references for carbon materials, metal/metal oxide nanoparticles… and for methods.
o Page 7 line 268: add the reference for Ratiometric electrochemical sensors
o Summarized the strategies in preparation of microelectrodes in section 2.
o Ascorbic acid often coexists with dopamine in biological samples. Besides, the oxidation potential for ascorbic acid can be similar to dopamine. How to increase the selectivity of ascorbic acid to dopamine? Provide an explanation in part 2.2.
Author Response
Please, see the attachment.

Reviewer 2 Report
1. As we all know, the electrochemical sensing is a three-electrode system. How to realize the integration of carbon fiber and acupuncture needle working microelectrode with reference electrode and counter electrode in live brains?
2. There are many interferences coexisting in live brains and blood samples, how do these sensors (e-NCNF, CNTs, VACNT-CFs, SWCNTs/Nafion® modified CFME) overcome the interference of several electroactive species (metal ions, neurotransmitters, etc.)?
3. The nanoporous gold nanomaterials have catalytic properties for many small molecules,such as H2O2, dopamine, uric acid, nitrite, and so on, how to avoid their influence on AA detection?
4. The specific surface area of acupuncture needle is small, how to modify the active materials? How to modify enough active materials on it to meet the needs of detection?
5. Using microelectrode arrays is helpful in analyte sensing systems for diagnostic and therapeutic monitoring, What is the normal AA concentration range of our body? Which diseases are associated with abnormal AA levels? Is there specificity?
6. If the AA can be quickly excreted, how does the normal AA intake work in our body?
7. What is the difference of AA sensors between clinical, pharmaceutical, and food safety applications?
8. What are the major challenges of the Microsensors applied to quantify AA testifies in the food sample matrices?

Author Response
Please, see the attachment.
